# Loneliness and Its Associated Factors Nine Months after the COVID-19 Outbreak: A Cross-National Study

**DOI:** 10.3390/ijerph18062841

**Published:** 2021-03-11

**Authors:** Tore Bonsaksen, Mariyana Schoultz, Hilde Thygesen, Mary Ruffolo, Daicia Price, Janni Leung, Amy Østertun Geirdal

**Affiliations:** 1Department of Health and Nursing Science, Faculty of Social and Health Sciences, Inland Norway University of Applied Sciences, 2418 Elverum, Norway; 2Faculty of Health Studies, VID Specialized University, 4306 Sandnes, Norway; hilde.thygesen@vid.no; 3Department of Health and Life Sciences, Northumbria University, Newcastle upon Tyne NE1 8ST, UK; mariyana.schoultz@northumbria.ac.uk; 4Department of Occupational Therapy, Prosthetics and Orthotics, Faculty of Health Science, Oslo Metropolitan University, 0130 Oslo, Norway; 5School of Social Work, University of Michigan, Ann Arbor, MI 48109, USA; mruffolo@umich.edu (M.R.); daiciars@umich.edu (D.P.); 6Faculty of Health and Behavioural Science, University of Queensland, Brisbane, QLD 4072, Australia; j.leung1@uq.edu.au; 7Department of Social Work, Faculty of Social Sciences, Oslo Metropolitan University, 0130 Oslo, Norway; amyoge@oslomet.no

**Keywords:** concerns, coronavirus, cross-national study, pandemic, social distancing, social media

## Abstract

COVID-19 has been a global healthcare concern impacting multiple aspects of individual and community wellness. As one moves forward with different methods to reduce the infection and mortality rates, it is critical to continue to study the impact that national and local “social distancing” policies have on the daily lives of individuals. The aim of this study was to examine loneliness in relation to risk assessment, measures taken against risks, concerns, and social media use, while adjusting for sociodemographic variables. The cross-sectional study collected data from 3474 individuals from the USA, the UK, Norway, and Australia. Loneliness was measured with the de Jong Gierveld Loneliness Scale. Multiple linear regression was used in the analysis of associations between variables. The results showed that concerns about finances were more strongly associated with social loneliness, while concerns about the future was more strongly associated with emotional loneliness. Longer daily time spent on social media was associated with higher emotional loneliness. In conclusion, pandemic-related concerns seem to affect perceptions of loneliness. While social media can be used productively to maintain relationships, and thereby prevent loneliness, excessive use may be counterproductive.

## 1. Introduction

Loneliness has been understood as comprising three essential elements [1]: First, a perceived lack or deficiency in a person’s social network indicates that relationships with other people are scarce; second, loneliness is a subjective experience, which means that it cannot be determined by others’ observation or with reference to universal standards; third, the loneliness experience is unpleasant and distressing. Thus, loneliness focuses on lack of relationships with other people and the emotional distress caused by lack of connection.

At the time of the COVID-19 outbreak, “social distancing” became the main policy for public behavior [2]. This implied maintaining an appropriate distance from people outside your immediate household. Moreover, people were encouraged to stay at home in relative isolation to prevent viral spread, and non-essential businesses were temporarily closed [3], leading to a sharp increase in unemployment rates internationally [4]. The restrictive social distancing policies and the general sense of uncertainty during the COVID-19 outbreak ultimately reduced contact between people and instigated a growing worldwide concern about increased loneliness [5,6,7,8] and more mental health problems in the general population [9,10,11,12,13].

Some population groups may be at higher risk of experiencing loneliness during the pandemic. For example, older people have been considered at risk of experiencing loneliness and subsequent mental health problems [14,15,16,17,18]. During the pandemic, higher age together with underlying chronic health conditions are considered the most prominent risk factors for complications and death [19]. This has possibly encouraged older adults to be extra careful with their social activities and resulted in them becoming particularly vulnerable to loneliness. However, studies conducted during the pandemic have also found increased levels of loneliness among young adults, with increases in depression largely explained by the increase in loneliness [20]. Moreover, a recent study from the early stage of the COVID-19 outbreak also found that unemployed persons reported more loneliness and poorer mental health and quality of life, compared to those who were employed [21].

The pandemic has brought on many worries and uncertainties related to job security, finances, health, and vaccines. With people losing their jobs or facing prospects of job losses, many are confronted with financial insecurities. Worries about finances have been shown to be associated with stress symptom levels compatible with post-traumatic stress disorder (PTSD) [9]. Moreover, witnessing sporadic increases in the number of COVID-19 cases, even under strict social distancing regimens, is potentially increasing people’s concerns about their own health, or the health of their family and close ones. While vaccines have been distributed since early 2021, mass vaccination takes time and will not likely resolve the situation instantly. Thus, the duration of the pandemic and the prolonged uncertainty about when life can get back to normal further contribute to the sense of meaninglessness and concern about the future [22] and therefore promote a further increase in loneliness.

While national and international health authorities are keeping the public informed about the pandemic situation daily, social media as powerful information agents across the world are broadcasting a mix of objective and biased information about the pandemic, its origins, and effects. Reliance on information posted on social media may be risky, and people could perceive posts and debates on social media as polarizing or even distressing [23,24]. Recent studies investigating social media use in relation to mental health have found that increased use of social media is associated with poorer mental health [25,26,27]. However, the relationship between social media use and loneliness may vary according to age group, as found in a study from the early stage of the COVID-19 outbreak [28].

COVID-19 has been a global health care concern that impacts multiple aspects of individual and community wellness. As one moves forward with different methods to reduce the infection and mortality rates, it is critical to continue to study the impact that national and local “social distancing” policies have on the daily lives of individuals. This study is a follow-up on the previous data collected during the first 90 days of the COVID-19 pandemic and reports on loneliness and its associated factors among adults in a cross-national sample nine months after the outbreak.

### Study Aim

The aim of this study was to examine loneliness in association with pandemic-related concerns, risk assessment, measures taken against risks, and social media use nine months after the COVID-19 pandemic emerged, while adjusting for sociodemographic variables.

## 2. Materials and Methods

### 2.1. Design and Procedures

The study had a cross-sectional survey design. The link to the survey was distributed through social media in each of the involved countries between 24 October and 29 November 2020. A landing site for the survey was established at the researchers’ universities; OsloMet-Oslo Metropolitan University, Norway; University of Michigan, USA; Northumbria University, UK; and the University of Queensland, Australia. The initiator of the project was AØG from OsloMet. Due to ethical considerations and permissions in each of the countries, each country had their own project lead. The survey was simultaneously co-developed by the researchers in two languages, Norwegian and English, and was based on a previous survey conducted by the research group in the early phase (April 2020) of the pandemic outbreak [21,27]. Language and cultural differences were considered during the survey development process. This means that the Norwegian phrasing of each item would convey the same meaning content as the corresponding English item, while considering the different grammatical structures and nuances in the culturally embedded meaning of words allowed us to use the phrase that would most effectively convey the meaning in each of the languages.

### 2.2. Inclusion and Exclusion

To be included in the study, participants had to be 18 years or older, understand Norwegian or English and live in Norway, the USA, the UK, or Australia, and be able to access the electronic survey.

### 2.3. Measures

#### 2.3.1. Sociodemographic Characteristics

Sociodemographic variables included age group (18–29 years, 30–39 years, 40–49 years, 50–59 years, 60–69 years, 70 years and above), gender identity (male, female, other, prefer not to respond), highest completed education level (high school or associated/technical degree or lower, bachelor’s degree, master’s/doctoral degree), cohabitation (living with a spouse or partner, or not), and employment status (having full-time or part-time employment, versus not).

#### 2.3.2. Pandemic-Related Concerns

Pandemic-related concerns were assessed with four separate items, related to health, finances, next of kin, and the future, respectively. All items were phrased: “During the ongoing COVID-19 pandemic, are you worried about…” followed by “your own health”, “your own financial situation”, “your next of kin” or “the future”. Response options were on a 0–4 rating scale, indicating totally disagree (0), disagree (1), neither agree or disagree (2), agree (3), and totally agree (4).

#### 2.3.3. Risk Assessment

Risk assessment was measured with two separate items. The participants were asked to respond to the question: “I consider myself to be at high risk of a fatal outcome if I tested positive for the COVID-19 virus”, with response options being no (0), maybe (1), and yes (2). The second question asked: “Where I currently live, the rate of COVID-19 infection is…”, with response options being very low (1), low (2), neither high nor low (3), high (4), and very high (5).

#### 2.3.4. Measures Taken against Risks

The participants were asked whether they had been in quarantine, either due to the authorities’ regulations or due to having had close contact with a person who had tested positive for the coronavirus. Participants who responded “yes” to either or both questions were classified as having been in quarantine. The participants were also asked whether they were or had been in self-isolation due to their own or family members’ risk of complications if infected. In this study, responses to this question were re-coded to represent yes versus no. Non-response to any of these questions was interpreted as not having been in quarantine or self-isolation, respectively.

#### 2.3.5. Social Media Use

The participants were asked to indicate the amount of time they had spent on social media on a typical day during the last month. In line with the work of Ellison and co-workers [29], response options were less than 10 min, 10–30 min, 31–60 min, 1–2 h, 2–3 h, and more than 3 h.

#### 2.3.6. Loneliness

The Loneliness Scale [30] comprises six statements, each rated on a discrete scale from 0 (totally disagree) to 4 (totally agree). It measures two aspects of loneliness, namely social loneliness (e.g., “There are plenty of people I can rely on when I have problems”) and “emotional loneliness” (e.g., “I experience a general sense of emptiness”). Prior studies have found a two-factor solution to be the best fit, and that the items should therefore be treated as constituting two different scales reflecting the social and emotional aspects of loneliness, respectively [30,31]. For both scales, the score range is 0–12, with higher scores indicating more loneliness. However, an overall measure of loneliness is also often established by combining all six items in one scale (score range 0–24). Cronbach’s α in this study was 0.88 for social loneliness, 0.70 for emotional loneliness, and 0.80 for overall loneliness.

### 2.4. Statistical Analysis

Overall, social and emotional loneliness (means and standard deviations) were calculated for each category of the independent variables: age group, sex, education level, cohabitation, and employment status. Depending on the number of group categories, group differences were examined using one-way analysis of variance (ANOVA) and the independent *t*-test. Pearson’s correlation coefficient *r* was used to assess the strength of the crude associations between each of the independent variables and the outcomes (social, emotional, and overall loneliness). Variables significantly associated with the at least one of the outcomes in the unadjusted analysis were included in the subsequent multivariate analyses. Multiple linear regression analyses were used to assess associations between each of the independent variables and the outcomes, while adjusting for all included variables. Variables were entered in five steps, representing sociodemographic variables, concerns, risk assessment, measures taken against risk, and social media use. Specifically, the regression model was constructed as follows: (1) age group, gender, education level, relationship status, and employment; (2) concerns about health, concerns about finances, concerns about next of kin, and concerns about the future; (3) assessment of own health risk if infected and perceived rate of infection in the area of living; (4) having been in quarantine, and having been in self-isolation; and (5) time spent on social media daily during the last month. Standardized beta weights (*β*) were reported as effect size, and according to Cohen [32], effect sizes about 0.10 were interpreted as small, effect sizes about 0.30 as moderate, and effect sizes about 0.50 as large. The outcome variance proportions explained by the models were reported. Statistical significance was set at *p* < 0.01, due to the large sample size. Missing values were handled by case-wise deletion, and actual *n* is reported for all analyses.

### 2.5. Ethics

The data collected in this study were anonymous. The researchers adhered to all relevant regulations in their respective countries concerning ethics and data protection. The study was approved by OsloMet (20/03676) and the regional committees for medical and health research ethics (REK; ref. 132066) in Norway, reviewed by the University of Michigan Institutional Review Board for Health Sciences and Behavioral Sciences (IRB HSBS) and designated as exempt (HUM00180296) in USA, by Northumbria University Health Research Ethics (HSR1920-080) in UK, and (HSR1920-080 2020000956) in Australia.

## 3. Results

### 3.1. Participants

The sample comprised 3474 individuals from Norway (*n* = 547, 15.7%), the USA (*n* = 2130, 61.3%), the UK (*n* = 640, 18.4%), and Australia (*n* = 157, 4.5%). The number of participants was relatively similar across age groups, with declining numbers in the age groups above 70 years. The majority were women (73.3% women versus 22.2% men), with 48 (1.4%) participants reporting gender identity to be “other” and 36 (1.0%) preferring not to report their gender. Seventy-one percent had education at the bachelor’s degree level or higher. More than half of the sample lived with a spouse or partner (58.7%), while full-time or part-time employment was held among 66.3%.

### 3.2. Loneliness in Sample Subgroups

Table 1 displays the levels of social, emotional, and overall loneliness according to sample subgroups. Emotional and overall loneliness were higher among the younger age groups. Men reported more social loneliness than women, while women reported more emotional loneliness than men. Participants with lower levels of education reported more loneliness compared to those with higher education levels, while those living with a spouse or partner reported less loneliness than their counterparts. Participants with employment reported lower levels of social and overall loneliness, compared to those not in employment.

### 3.3. Associations with Loneliness

Crude (unadjusted) associations with loneliness are displayed in Table 2. Most associations were statistically significant. Concerns about finances had a moderately strong association with emotional loneliness (*r* = 0.37, *p* < 0.001) and with overall loneliness (*r* = 0.39, *p* < 0.001). Concerns about the future bordered toward a strong relationship with emotional loneliness (*r* = 0.47, *p* < 0.001) and with overall loneliness (*r* = 0.42, *p* < 0.001). More social media use was associated with higher emotional loneliness (*r* = 0.23, *p* < 0.001).

The results from the multiple linear regression analyses are reported in Table 3. Adjusted by all included variables, higher levels of social loneliness were associated with concerns about finances (*β* = 0.21, *p* < 0.001) and the future (*β* = 0.15, *p* < 0.001). Higher levels of emotional loneliness were associated with concerns about health (*β* = 0.07, *p* < 0.001), finances (*β* = 0.13, *p* < 0.001), and the future (*β* = 0.31, *p* < 0.001). Higher emotional loneliness was also associated with more time spent on social media (*β* = 0.07, *p* < 0.001). Higher levels of overall loneliness were associated with concerns about finances (*β* = 0.20, *p* < 0.001) and the future (*β* = 0.27, *p* < 0.001), and with more social media use (*β* = 0.05, *p* < 0.01). Among the sociodemographic variables, cohabitation and higher education levels were relatively consistently associated with lower levels of loneliness. Higher age was related to higher levels of social loneliness (*β* = 0.08, *p* < 0.01) and with lower levels of emotional loneliness (*β* = −0.15, *p* < 0.001).

## 4. Discussion

### 4.1. Factors Associated with Loneliness

Pandemic-related concerns, in particular concerns about finances and the future, were associated with loneliness. Previous research on loneliness has also suggested that those with a low household income experience more loneliness, compared to those with high household income [33]. Bu and colleagues [33] noted that while this was the case before the pandemic outbreak (based on data collected in 2017–2019), new data collected between March and May 2020 suggest that this relationship was even stronger during the early stage of the pandemic. As our study showed no relationship between employment and loneliness, it appears that the relationship between financial concerns and loneliness is independent of current employment status. Interestingly, our study showed that concerns about finances were more strongly related to social loneliness, compared to emotional loneliness. During normal circumstances, money plays a crucial role in a person’s ability to participate in social arenas in the community (e.g., cafes, cinemas, concerts). While the opportunity to use such commercial social arenas was drastically reduced due to social distancing policies in the early stage of the pandemic, it may be that people are—nine months later—again inclined to perceive money as an important means to access social relationships. However, social relationships during the pandemic are often maintained by one’s virtual rather than physical presence.

Conversely, concerns about the future were more strongly related to emotional loneliness, compared to social loneliness. It is possible that such concerns—abstract and perhaps vague concerns about an unknown future, as opposed to practical concerns of managing everyday life at the present—are difficult to discuss over the phone or on video calls, even with friends and family. If so, they may tend to become private concerns and can as such be interpreted as possible precursors of emotional loneliness. Concerns about the future may be related to existential questions about purpose and meaning in life, and for some individuals, the pandemic situation appears to evoke feelings of emptiness and being remote from the world, as suggested previously [22].

Social media use was found to be significantly, but weakly, related to higher levels of emotional (and overall) loneliness. Thus, while more time spent on social media did not affect levels of social loneliness, more time spent on social media correlated with slightly higher levels of emotional loneliness, implying stronger feelings of emptiness, rejection, and remoteness from people. Social media have gained enormous popularity since they emerged [34,35] and have been used increasingly during the pandemic [36]. However, our findings are in line with studies indicating that their use may instigate more, rather than less, loneliness [36]. Social media use has also been associated with poorer mental health [25,26,27]. Reciprocal relationships are equally possible—more loneliness may increase social media use, whereas increased social media use in turn may increase loneliness. Serving as a way to stay connected, social media may also be a reminder of the uncertainties present with the pandemic and may therefore increase feelings of loneliness instead of alleviating them.

On the other hand, some researchers have argued that psychological outcomes related to social media use may not only be concerned with the amount of time spent on social media, but also with the motives for their use [37]. While pro-social motives, such as having contact with friends and family, may reduce loneliness over time, compensation motives (using social media to compensate for lacking social skills in real life) or addiction motives (unable to log off) may increase it [37,38].

Among the sociodemographic covariates, higher age was found to be related to higher social loneliness but lower emotional loneliness. Reduced size and quality of social networks is commonly found among persons in the older age groups [39], and people of older age may be less able to use digital tools as an alternative to direct contact in the COVID-19 era with the social distancing and sheltering at home policies in place [40]. In this situation, social media have become more important for maintaining social contacts [27]. A clinical trial that trained older adults to use social networking sites found that those who had participated in the training were more likely to use social networking and reported reduced feelings of being left out [41]. Thus, public health strategies to improve social media and digital communication literacy for older adults in the community may be recommended to reduce social loneliness in the older population.

It may be that older adults experience emotional loneliness at levels similar to those of younger people but are less likely to express it in surveys. Alternatively, older adults may be less prone to experience emotional loneliness. Life experience may buffer against feelings of emptiness and loss of meaning, despite having fewer people around in their daily life. Studies demonstrating less depression [42] and better global health [43] among those of older age indicate that life experience is a valuable resource that can buffer against emotional loneliness as an effect of the inevitable burdens that are introduced in older age. Still, the higher social loneliness in older adults is of concern. Loneliness in old age is generally acknowledged as an urgent public health problem [44], particularly in those living without a spouse or partner. In turn, during the COVID-19 pandemic, loneliness was found to be associated with higher levels of psychological distress [12] and even with malnutrition [45].

### 4.2. Implications

Many are likely to have concerns during the COVID-19 pandemic situation. This study implies that different types of concerns may be linked with different aspects of loneliness. Thus, knowledge about the kinds and levels of concerns people have may provide some indication about their susceptibility to social or emotional loneliness. Bluntly put, while people with predominantly financial concerns during the pandemic may be missing the opportunity to socialize with others, people who are concerned about the future may feel rather emotionally cut off from the world. Similarly, knowledge of the time people spend on social media on a daily basis may provide some indication about their susceptibility to loneliness. While concerns and feelings of loneliness may be viewed as natural responses during the current crisis, one should strive to reduce people’s burden by identifying those groups who may be most in need of support. This study has identified people with pandemic-related concerns and people with higher levels of social media use as having a higher risk of experiencing loneliness. These factors may therefore be considered during the planning and provision of mental health support in the communities.

### 4.3. Study Limitations

The study used a variety of measures, both well-tested questionnaires and measures that were developed by the researchers for this particular study. While the newly developed measures appear to be relatively straightforward and easy to understand (e.g., the questions about concerns, risk assessment, and measures taken against risk), their status as new and untested measures should be taken into consideration when interpreting the results of the study.

Respondents received invitations to participate through social media. With social media being an aspect for individuals to potentially engage with others, the responses are not inclusive of individuals that do not utilize social media and limits the ability to generalize the results to the general populations of the respective countries. The sample had a large proportion of women and persons with higher levels of education. Thus, also for these reasons, the sample should be considered skewed and not representative of the general population. Future studies may address this problem by using more sophisticated sampling methods, such as quota sampling or stratified sampling, as they may increase the chances of obtaining samples representative of the general population. Virtual university-sponsored research can unintentionally reach more college-educated respondents, as observed in our study, and reduce the diversity of age, race, gender identity, and socioeconomic status. Risk assessment allowed interpretation of the individual as they self-administered the tool. The study does not have pre-pandemic data to compare levels of loneliness before and after the outbreak. This study aims to explore loneliness and its associated factors nine months after the outbreak using cross-sectional data. A strength of the study is the relatively large and cross-nationally composed sample. Cross-national research engagement is known to reduce demographic bias with diverse geographical representation [46].

## 5. Conclusions

While risk and measures taken against risk were not related to loneliness, the results suggest that financial concerns are more strongly related to social loneliness, while concerns about the future are more strongly related to emotional loneliness. More time spent on social media was related to higher emotional loneliness. Thus, the degree to which people have pandemic-related concerns has bearings for their perceptions of loneliness. While social media can be used productively to maintain relationships, and thereby prevent loneliness, excessive use may be counterproductive. With vaccination still in its early days and new virus mutations surfacing, the future course of the pandemic is difficult to predict. Future studies are therefore needed to assess changes in loneliness over time.

## Figures and Tables

**Table 1 ijerph-18-02841-t001:** Social, emotional, and overall loneliness in sample subgroups.

Characteristics	Social Loneliness			Emotional Loneliness			Overall Loneliness		
	*n*	*M* (*SD*)	*p*	*n*	*M* (*SD*)	*p*	*n*	*M* (*SD*)	*p*
Age group			ns.			<0.001			<0.001
18–29 years	632	4.3 (2.9)		629	7.3 (2.6)		628	11.7 (4.6)	
30–39 years	701	4.5 (3.1)		704	6.4 (2.7)		701	10.9 (5.0)	
40–49 years	556	4.6 (3.3)		557	5.9 (2.7)		554	10.5 (5.1)	
50–59 years	438	4.2 (3.2)		438	5.4 (2.9)		436	9.6 (5.0)	
60–69 years	435	4.8 (3.0)		434	5.2 (2.9)		433	10.0 (4.9)	
70 years +	280	4.2 (2.9)		271	4.5 (2.8)		269	8.7 (4.8)	
Gender identity			<0.01			<0.001			0.17
Male	699	4.7 (3.2)		695	5.5 (3.2)		693	10.2 (5.3)	
Female	2304	4.3 (3.1)		2298	6.2 (2.7)		2288	10.5 (4.9)	
Education level			<0.001			<0.001			<0.001
High school/technical degree or lower	833	4.9 (3.2)		832	6.2 (3.0)		825	11.1 (5.0)	
Bachelor’s degree	1096	4.5 (3.1)		1091	6.2 (2.9)		1088	10.7 (5.0)	
Master’s/doctoral degree	1144	4.1 (3.0)		1140	5.7 (2.7)		1138	9.7 (4.9)	
Cohabitation			<0.001			<0.001			<0.001
Yes	1877	4.1 (3.0)		1873	5.6 (2.7)		1867	9.7 (4.8)	
No	1197	5.0 (3.2)		1191	6.6 (3.0)		1185	11.6 (5.1)	
Employment			<0.001			ns.			<0.01
Full-time or part-time	2091	4.2 (3.1)		2092	6.1 (2.8)		2083	10.3 (4.9)	
No employment	973	4.9 (3.1)		962	6.0 (3.0)		959	10.8 (5.1)	

Statistical tests are one-way ANOVA *F*-test (age groups and education level) and independent *t*-tests (all other variables). Cohabitation refers to “living with spouse or partner”.

**Table 2 ijerph-18-02841-t002:** Unadjusted (crude) associations between the independent variables and social, emotional, and overall loneliness.

Independent Variables	Social Loneliness		Emotional Loneliness		Overall Loneliness	
	*n*	*r*	*n*	*r*	*n*	*r*
Higher age	3042	0.00	3033	−0.30 ***	3021	−0.17 ***
Female gender	3003	−0.06 **	2993	0.10 ***	2981	0.03
Higher education level	3073	−0.11 ***	3063	−0.08 ***	3051	−0.11 ***
Living with spouse/partner	3074	−0.14 ***	3064	−0.17 ***	3052	−0.18 ***
Having employment	3064	−0.09 ***	3054	0.01	3042	−0.05 **
Concerned about health	3024	0.14 ***	3014	0.28 ***	3002	0.25 ***
Concerned about finances	3037	0.29 ***	3028	0.37 ***	3016	0.39 ***
Concerned about next of kin	3053	0.11 ***	3043	0.26 ***	3031	0.22 ***
Concerned about the future	3058	0.25 ***	3052	0.47 ***	3040	0.42 ***
Self-perceived high risk of complications	3072	0.10 ***	3062	0.00	3050	0.06 **
Higher infection rate in the living area	3072	0.03	3062	0.17 ***	3050	0.11 ***
Have been in quarantine	3074	0.06 **	3064	0.14 ***	3052	0.12 ***
Have been in self−isolation	3074	0.08 ***	3064	0.10 ***	3052	0.10 ***
More time spent on social media	2955	0.07 ***	2946	0.23 ***	2935	0.18 ***

Table content is Pearson’s *r*. ** *p*< 0.01, *** *p*< 0.001.

**Table 3 ijerph-18-02841-t003:** Multiple linear regression analysis showing adjusted associations with social, emotional, and overall loneliness.

Independent Variables	Social Loneliness (*n* = 2770)	Emotional Loneliness (*n* = 2766)	Overall Loneliness (*n* = 2755)
Sociodemographic variables	*β*	*β*	*β*
Higher age	0.08 **	−0.15 ***	−0.02
Female gender	−0.05 **	0.05 **	−0.01
Higher education level	−0.06 **	−0.04	−0.06 **
Living with spouse/partner	−0.12 ***	−0.11 ***	−0.14 ***
Having employment	−0.02	−0.04	−0.04
**R^2^ change**	**3.7%**	**12.1%**	**7.4%**
*Concerns*			
Concerns about health	0.02	0.07 **	0.05
Concerns about finances	0.21 ***	0.13 ***	0.20 ***
Concerns about next of kin	−0.06	−0.02	−0.05
Concerns about the future	0.15 ***	0.31 ***	0.27 ***
**R^2^ change**	**8.7%**	**16.1%**	**16.6%**
*Risk assessment*			
Self−perceived high risk of complications	−0.02	0.04	0.01
Higher infection rate in the living area	−0.02	−0.04	−0.03
**R^2^ change**	**0.1%**	**0.2%**	**0.1%**
*Measures taken against risk*			
Have been in quarantine	0.01	0.01	0.01
Have been in self−isolation	0.03	−0.00	0.01
**R^2^ change**	**0.1%**	**0.0%**	**0.0%**
*Social media use*			
More time spent on social media	0.02	0.07 ***	0.05 **
**R^2^ change**	**0.0%**	**0.4%**	**0.3%**
**Total explained outcome variance**	**12.6%**	**28.8%**	**24.4%**

Table content is standardized *β* weights. ** *p* < 0.01, *** *p* < 0.001.

## Data Availability

The data presented in this study are available on request from the corresponding author by completion of the research project. The data are not publicly available due to ongoing publication from the project and different data protection regulations in the four involved countries.

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
