# Peer review of "Loneliness and Its Associated Factors Nine Months after the COVID-19 Outbreak: A Cross-National Study"

_ijerph, 2021, doi:10.3390/ijerph18062841_

Round 1
Reviewer 1 Report
Dear researchers and authors of this article, thank you for sending me this interesting manuscript to review. This an interesting article and timely article as the pandemic holds. However, the manuscript potentially merits publication if certain important issues relating to the methodology and discussion section are further developed and explained.
Some points reported below
Page 3, line 110 the authors stated that language and cultural differences were considered during the survey development process. What does this mean? Needs clarification.
How the authors perform the translation of the instruments?
The authors performed a cross-sectional study. These types of studies have been now under consideration relating to the robustness of their results. Do the authors consider analyzing the data based on the quota sampling method to reduced potential biases?
Most of the instruments used here are in-house questionnaires so the validity of the results must be treated with caution.
Also, data before the pandemic outbreak is lacking. Are they any data that could be used in this direction?
Limitations of the study should be more carefully addressed.
Overall, the discussion would significantly benefit from further elaboration than merely reiterating the findings section. In the discussion section, the researcher should attempt to understand what the findings mean … and to discuss what the findings imply for society.
Reviewer 2 Report
Thank you for the opportunity to review this article titled “Loneliness and Its Associated Factor Nine Months After The COVID-10 outbreak: A Cross-National study”.
The aim of this project is to help understand the factors associated to loneliness during the COVID-19 pandemic. The authors obtained information of 3469 individuals from three different countries. In my opinion, having so many participants is the most important thing of all. It is an even more worthy outcome considering the current situation. The manuscript is well written and the methods are clearly explained. The questionnaires used to assess sociodemographic characteristics, pandemic-related concerns, risk assessment, measures taken against risks, social media uses and loneliness are complete enough to provide an overview of the current situation.
I really enjoyed the paper and found it very interesting.
Minor comments
- Which social media did you include? Any social media?
- Did you differentiate between the different social media? It would be interesting to add some comments about the different social media and their effect that each of one (or at least the most commonly used) have on individuals. For example, Twitter is used for different purposes than Instagram. In addition, social media more based on photos (e.g. Instagram) have different effects than social media based on text (e.g Twitter).
- Did the time spent in social media change overtime? Was is the same at the beginning of the pandemic that at the time of your study?
Round 2
Reviewer 1 Report
Many thanks for revising this article so carefully. No further remarks.